# Puerarin Attenuates Oxidative Stress and Ferroptosis via AMPK/PGC1α/Nrf2 Pathway after Subarachnoid Hemorrhage in Rats

**DOI:** 10.3390/antiox11071259

**Published:** 2022-06-27

**Authors:** Yi Huang, Honggang Wu, Yongmei Hu, Chenhui Zhou, Jiawei Wu, Yiwen Wu, Haifeng Wang, Cameron Lenahan, Lei Huang, Sheng Nie, Xiang Gao, Jie Sun

**Affiliations:** 1Department of Neurosurgery, Ningbo First Hospital, Ningbo Hospital, Zhejiang University School of Medicine, Ningbo 315010, China; huangy102@gmail.com (Y.H.); zhouchenhui1989@126.com (C.Z.); 21818268@zju.edu.cn (J.W.); wuyiwen@zju.edu.cn (Y.W.); 18857496593@139.com (H.W.); niesheng@163.com (S.N.); 2Department of Physiology and Pharmacology, Loma Linda University, Loma Linda, CA 92350, USA; honggangsiq@163.com (H.W.); yongmeihu2019@gmail.com (Y.H.); lhuang@llu.edu (L.H.); 3Key Laboratory of Precision Medicine for Atherosclerotic Diseases of Zhejiang Province, Ningbo 315010, China; 4Department of Neurosurgery, People’s Hospital of Leshan, Leshan 614099, China; 5Department of Nursing, Henan Provincial People’s Hospital, Zhengzhou 450003, China; 6Burrell College of Osteopathic Medicine, Las Cruces, NM 88001, USA; cameron.lenahan@burrell.edu

**Keywords:** subarachnoid hemorrhage, early brain injury, puerarin, oxidative stress, ferroptosis

## Abstract

Puerarin was shown to exert anti-oxidative and anti-ferroptosis effects in multiple diseases. The goal of this study was to explore the neuroprotective effect of puerarin on early brain injury (EBI) after subarachnoid hemorrhage (SAH) in rats. A total of 177 adult male Sprague Dawley rats were used. SAH was included via endovascular perforation. Intranasal puerarin or intracerebroventricular dorsomorphin (AMPK inhibitor) and SR18292 (PGC1α inhibitor) were administered. The protein levels of pAMPK, PGC1α, Nrf2, 4HNE, HO1, MDA, ACSL4, GSSG, and iron concentration in the ipsilateral hemisphere were significantly increased, whereas SOD, GPX4, and GSH were decreased at 24 h after SAH. Moreover, puerarin treatment significantly increased the protein levels of pAMPK, PGC1α, Nrf2, HO1, SOD, GPX4, and GSH, but decreased the levels of 4HNE, MDA, ACSL4, GSSG, and iron concentration in the ipsilateral hemisphere at 24 h after SAH. Dorsomorphin or SR18292 partially abolished the beneficial effects of puerarin exerted on neurological dysfunction, oxidative stress injury, and ferroptosis. In conclusion, puerarin improved neurobehavioral impairments and attenuated oxidative-stress-induced brain ferroptosis after SAH in rats. The neuroprotection acted through the activation of the AMPK/PGC1α/Nrf2-signaling pathway. Thus, puerarin may serve as new therapeutics against EBI in SAH patients.

## 1. Introduction

Subarachnoid hemorrhage (SAH) is an acute hemorrhagic cerebrovascular disease in which blood flows into the subarachnoid space after a cerebral blood vessel rupture [1]. SAH accounts for about 10% of acute strokes and is associated with a high mortality and disability rate [2]. Despite recent progress in the clinical diagnosis and surgical approach, the lack of effective pharmacological treatment renders poor prognosis in SAH patients.

Early brain injury (EBI), occurring within 72 h of SAH, is a main pathological process in contribution to the high mortality and severe complications after SAH [3]. Suzuki et al. [4] showed that iron content in the cerebrospinal fluid and cortex was increased on days 3 and 4 after SAH in patients. The blood degradation products of the subarachnoid space can activate the oxidation reaction and lipid peroxidation, which leads to the early ferroptosis after SAH [5]. Ferroptosis is an iron-dependent cell death, which is connected with oxidative stress and played key roles in the pathogenesis of stroke and ischemia–reperfusion injury [6]. Accumulating evidence suggests that ferroptosis played an important role in the EBI after SAH [7]. Therefore, early interventions to minimize oxidation reactions and ferroptosis may be effective in treating SAH.

Puerarin is a type of flavonoid glycoside extracted from the Pueraria lobata root [8]. Studies have shown that puerarin has neuroprotective effects in several central nervous system diseases, including Parkinson’s disease [9], Alzheimer’s disease [10], acute spinal cord injury [11], brain ischemia/reperfusion injury [12,13], and intracerebral hemorrhage [14]. As an antioxidant to protect the cells against apoptosis caused by oxidative stress [15], Puerarin was shown to preserve the activities of antioxidant enzymes [16,17]. It modulates oxidative stress and mitochondrial function via 5′ adenosine monophosphate-activated protein kinase (AMPK) [18] and Nrf2 [19]. The activations of AMPK and Nrf2 played critical roles in modulating effects on ferroptosis [20,21]. Phosphorylation of PGC1α is a key indicator of activation of the AMPK-signaling pathway, and acts as a bridge to connect the two signaling pathways of AMPK and Nrf2 [22]. Recent studies showed that puerarin attenuated ferroptosis in heart failure [23] and lung injury diseases [24]. However, the roles of puerarin on ferroptosis in the setting of SAH remain unknown.

In light of previous studies, we hypothesized that puerarin would attenuate oxidative stress and ferroptosis of EBI in part through the AMPK/PGC1α/Nrf2 pathway after SAH in rats.

## 2. Materials and Methods

### 2.1. Animals

All experimental procedures were approved by the Institutional Animal Care and Use Committee (IACUC) of Loma Linda University (LLU, No. 8170018) and Ningbo University (NBU, No. NBU20210032). Adult male Sprague Dawley (SD) rats (weight 280–300 g) were housed in the Animal Care Facility of LLU and NBU. Rats were randomly divided into Sham and SAH groups.

### 2.2. SAH Model

The modified endovascular perforation was performed to induce SAH in rats as previously described [25]. Briefly, under 3% isoflurane anesthesia, the rats were intubated and connected to the ventilator (Harvard Apparatus, Holliston, MA, USA) in a supine position. The left common carotid artery, internal carotid artery, and external carotid artery were exposed and a monofilament nylon thread (4–0) with sharpened tip was inserted into the middle cerebral artery through the left branch, and the artery was further carefully punctured. The same procedure was performed in sham-operated rats without blood vessel perforation. After the completion of the operation, the rat was extubated and kept in a separate warm cage until it recovered from anesthesia. Two independent researchers performed SAH grading after the rats were sacrificed. Referring to Sugawara’s study [26], the brain was organized into 6 regions and scored from 0 to 3 according to the amount of blood covering the surface of each region, and the total SAH grade score was the sum of the scores for each region (0–18).

### 2.3. Drug Administration

Puerarin (P5555, Sigma, St. Louis, MO, USA) was diluted in phosphate-buffered saline (PBS, 0.01 M, Vehicle 1) and administered via intranasal (i.n) route at 1 h after SAH as previously described [27]. Briefly, rats were anesthetized and placed in the supine position. A total volume of 20 μL of vehicle or three different concentrations of puerarin (20 mg/kg, 40 mg/kg and 80 mg/kg) were slowly injected with 5 μL/one nostril and alternated every 5 min.

An AMPK inhibitor Dorsomorphin (P5499, Sigma-Aldrich, St. Louis, MO, USA) was diluted in 20% dimethyl sulfoxide (DMSO) in PBS (Vehicle 2). A PGC1α inhibitor SR18292 (SML1246, Sigma-Aldrich, MO, USA) was dissolved in 10% DMSO and 10% Tween 80 in sterile saline (Vehicle 3). Dorsomorphin (0.1 μmol, 10 μL) [28], SR18292 (2.5 μg/μL, 10 μL) [29], and vehicles (10 μL) were administered intracerebroventricularly (i.c.v) at 1 h before SAH. Briefly, the isoflurane-anesthetized rats were placed in a stereotaxic apparatus. After a burr hole was drilled, a precision 10 μL syringe (Hamilton Company, Reno, NV, USA) was inserted at 1.0 mm caudal and 1.5 mm lateral to the bregma, as well as 3.5 mm in depth into the left lateral ventricle. The drugs were slowly administered using an infusion pump (Quintessential Stereotaxic Injector, Stoelting Co., Wood Dale, IL, USA) at a speed of 0.5 μL/min.

### 2.4. Experimental Group

The experimental design and animal groups are shown in Figure 1. Rats were randomly assigned to four separate experimental groups. The SAH grade scores greater than 9 were included in the study. Rats with SAH scores less than 8 or dead rats were excluded. The operations were repeated until sufficient sample size was reached for statistical analysis.

#### 2.4.1. Experiment 1. Time Course Study

To determine the endogenous protein levels of pAMPK and PGC1α in the Sham and 5 SAH groups, A total of 36 rats were randomly divided into Sham group (*n* = 6) and post-SAH groups of 3 h (*n* = 6), 6 h (*n* = 6), 12 h (*n* = 6), 24 h (*n* = 6), and 72 h (*n* = 6). Western blots were performed to determine the protein levels of pAMPK and PGC1α in the brain tissues of ipsilateral hemisphere.

#### 2.4.2. Experiment 2. Short-Term Outcome Study

To evaluate the effect of puerarin on short-term outcomes after SAH, 42 rats were randomly assigned into six groups: Sham (*n* = 10), SAH + Vehicle (PBS) (*n* = 10), SAH + Puerarin (20 mg/kg, *n* = 6), SAH + Puerarin (40 mg/kg, *n* = 10), and SAH + Puerarin (80 mg/kg, *n* = 6) groups [30,31]. Puerarin or vehicle were administered via i.n. at 1 h after SAH. Neurological test, Western blot, enzyme-linked immunosorbent assay (ELISA), immunofluorescence staining, and the terminal deoxynucleotidyl transferase dUTP nick end-labeling (TUNEL) assays were performed at 24 h after SAH.

#### 2.4.3. Experiment 3. Long-Term Outcome Study

To evaluate the effect of puerarin on long-term outcomes after SAH, 30 rats were randomly assigned into three groups (*n* = 10, per group): Sham, SAH + Vehicle, and SAH + Puerarin groups. Rotarod tests were performed on days 7, 14, and 21 after SAH. The Morris water maze test was performed on days 21–26 after SAH. Nissl staining was used to analyze neuronal degeneration in brain tissues at 26 days after SAH.

#### 2.4.4. Experiment 4. Mechanism Study

To explore the neuroprotective mechanism of puerarin, the selective AMPK inhibitor dorsomorphin or PGC1α inhibitor SR18292 was administered at 1 h before SAH. A total of 42 rats were randomly assigned into seven groups (*n* = 6, per group): Sham, SAH + vehicle 1, SAH + Puerarin, SAH + Puerarin +Vehicle 2, SAH + Puerarin + Dorsomorphin, SAH + Puerarin + Vehicle 3, and SAH + Puerarin + SR18292. Neurological tests, ELISA, and Western blots were performed at 24 h after SAH.

### 2.5. Neurological Score

The neurological scores were assessed using the modified Garcia test and beam balance test [25]. The Modified Garcia score (3–18) was the sum of 6 separate trial scores, including spontaneous activity (scores 0–3), symmetry (0–3), forelimb extension (0–3), climbing (1–3), body proprioception (1–3), and response to vibrators (1–3). Beam balance test was used to record the distance rats walked on a balance beam in a quiet environment, scoring 0–4. A higher score indicated a better performance on the neurobehavioral tests. The tests were performed at 24 h after SAH by two independent researchers blind to the experimental group information.

The rotarod test was used to detect the physical coordination of rats at 7, 14, and 21 days after SAH as previously described [25]. Briefly, the rats were placed on the rotarod apparatus (Columbus Instruments, Columbus, OH, USA) with an initial speed of 5 or 10 revolutions per minute (RPM) and an acceleration of 0.4 revolutions/s. Latencies to fall from the rotating device were recorded by researchers.

The Morris water maze test was performed to evaluate spatial-learning capacity from days 21–26 after SAH. The water maze was divided into four quadrants. Rats were placed in the pool from different directions and were tasked to seek a platform on the water within 1 min. Water maze training was conducted on days 1–5, and the water surface platform was removed for assessment on day 6. An automated video-tracking system (AVTAS v3.3; AniLab, Ningbo, China) was used to record the trajectory and time of the rat’s movement during the water maze.

### 2.6. Immunofluorescence Staining

The 8-hydroxy-2′-deoxyguanosine (8OHdG) and Mitosox staining were used to assess to the oxidative-stress-induced DNA damage and mitochondrial superoxide level [25]. Briefly, the freshly prepared frozen 15 μm coronal brain sections were immersed in antigen retrieval solution (PH 6.0) and heated in a microwave oven for 15 min. Sections were mounted with 3% H_2_O_2_ for 10 min to block endogenous peroxidase. The 8OHdG antibody (ab62623, Abcam, Waltham, MA, USA) and Mitosox antibody (M36008, Thermo Fisher, Waltham, MA, USA) were incubated at room temperature. The cell nuclei were stained by 4′,6-diamidino-2-phenylindole (DAPI, Sigma-Aldrich, St. Louis, MO, USA). Finally, the brain slices were observed using a fluorescence microscope (DMi8, Leica Microsystems, Wetzlar, Germany) under 400× or 200× magnification and averaged from four randomly selected regions within the ipsilateral entorhinal cortex.

### 2.7. Terminal Deoxynucleotidyl Transferase dUTP Nick End-Labeling (TUNEL) Staining

TUNEL staining was performed to evaluate neuronal cell death in brain tissue using In Situ Apoptosis Detection Kit (Roche, Wilmington, MA, USA). Briefly, brain sections were incubated with anti-NeuN antibody (1:200, ab104224, Abcam, USA) overnight at 4 °C. Next, the sections were treated with 50 μL of TUNEL reaction buffer, followed by incubation at 37 °C for 60 min in a dark and humidified atmosphere. The cell nuclei were stained through DAPI. Finally, the brain slices were observed using a fluorescence microscope under 200× magnification and were averaged from four randomly selected regions within ipsilateral entorhinal cortex.

### 2.8. Nissl Staining

Nissl staining was performed to evaluate hippocampal neuronal damage at 26 days after SAH [25]. Briefly, the brain tissue was cut into 15 μm coronal sections, immersed with 0.5% crystal violet, dehydrated with 100% alcohol, and transparentized with xylene. Finally, the slices were observed under a light microscope. The damaged neurons were counted within ipsilateral entorhinal cortex under 400× magnification and were averaged from four randomly selected regions.

### 2.9. Western Blot Analysis

Rat ipsilateral brain tissues were lysed and homogenized in RIPA buffer (Santa Cruz Biotechnology, Dallas, TX, USA) with protease and phosphate inhibitors, followed by centrifugation at 14,000 rpm at 4 °C for 30 min. The protein concentration was measured and samples were re-suspended in loading buffer and denatured at 95 °C for 10 min. Equal quantities of protein were loaded into sodium dodecyl sulfate polyacrylamide gel electrophoresis (SDS-PAGE) for protein separation, and then transferred to a nitrocellulose membrane (45 μm). The membranes were incubated with the primary antibody at 4 °C for 6–8 h. The primary antibodies used are listed as follows: anti-pAMPK (1:500, Cell Signaling Technology, Danvers, MA, USA), anti-PGC1α (1:1000, Cell Signaling Technology, USA), anti-Nrf2 (1:500, Abcam, USA), anti-4HNE (1:1000, Abcam, USA), anti-HO1 (1:30,000, Abcam, USA), anti-glutathione peroxidase 4 (GPX4) (1:3000, Proteintech, Wuhan, China), anti-Acyl-CoA synthetase long-chain family member 4 (ACSL4) (1:1000, Proteintech, Wuhan, China), and anti-β-actin (1:3000, Santa Cruz Biotechnology, Dallas, TX, USA). The appropriate secondary antibodies (1:3000, Santa Cruz Biotechnology, USA) were incubated at room temperature for 1 h. The immunoreactive bands were developed using an ECL Plus kit (Genesee Scientific, San Diego, CA, USA), exposed to X-ray film, and lastly, quantitatively analyzed using Image J software (NIH, Bethesda, MD, USA).

### 2.10. ELISA Assay

ELISA assay was performed to evaluate the levels of brain oxidative damage and ferroptosis indexes [32]. After cardiac perfusion, the ipsilateral brain tissues were homogenized with cold lysis buffer, and the supernatant was separated via centrifugation at 4 °C at 14,000 rpm for 30 min. The levels of malondialdehyde (MDA) (S0131), antioxidant enzyme superoxide dismutase (SOD) (S0101), and glutathione (GSH)/oxidized glutathione (GSSG) (S0053) were measured using ELISA assay kits according to the manufacturer’s instructions (Beyotime Biotechnology, Shanghai, China). Briefly, MDA detection was based on the chromogenic reaction between MDA and thiobarbituric acid, and measured at 535 nm light wave [33]. SOD analysis was performed according to the WST-8 method and detected at 450 nm light wave [34,35]. The GSH/GSSG assay was performed by glutathione reductase treatment and the concentration is detected at 412 nm light wave [33].

### 2.11. Brain Tissue Iron

Brain tissue iron concentration was measured as previously described [36]. Briefly, after cardiac perfusion with PBS (4 °C, 0.01 M, pH 7.4), the remaining subarachnoid blood was carefully removed from the basal surface of the brain. The ipsilateral brain tissues were homogenized with 1 mL of 8.5 mol/L HCl. After 60 min of hydrolysis at 90 °C, 20% trichloroacetic acid was added to precipitate the protein, and the supernatant was collected via centrifugation. We then washed the precipitate with 1 mL mixture of 4.25 mol/L HCl and 20% trichloroacetic acid (1:1), and the supernatant was collected via centrifugation. Next, 4 mL of 1 mol/L sodium citrate was added and the pH was adjusted to 3.1. Total brain iron content was assayed by a spectrophotometer with ferrozine as the color reagent.

### 2.12. Statistical Analysis

All statistical analyses were performed using GraphPad Prism 8 software (San Diego, CA, USA). The protein, immunostaining, Nissl staining, and neurological scores were firstly evaluated for the normality and homogeneity of variances using Shapiro–Wilk test and Levene’s test, respectively, and then were analyzed using one-way ANOVA followed by Tukey’s post hoc test. Rotarod and water maze tests were measured by two-way ANOVA Tukey’s post hoc test. The results are presented as mean ± standard deviation (SD), and *p* < 0.05 was considered statistically significant.

## 3. Results

### 3.1. Mortality and SAH Grades

As shown in Table 1, a total of 177 male rats were used in this present study. Among them, 145 rats underwent SAH surgery, whereas 32 rats underwent sham surgery. The overall mortality rates were 15.2% (22/145) and 0% (0/32) in the SAH and sham groups, respectively. A total of 27 SAH rats were excluded due to the mild SAH grade (SAH grade < 9, *n* = 5) or death during surgery (*n* = 22).

### 3.2. Puerarin Treatment Improved Short-Term Neurobehavioral Deficits at 24 h after SAH

As shown in Figure 2A, the blood clots were mainly distributed on the surface of the brain after SAH induction, but were rarely present in the sham group. No significant difference in SAH grade was found between all SAH groups (Figure 2B). The Western blot showed that endogenous expression of pAMPK and PGC1α started increasing at 3 h after SAH and peaked at 24 h (SAH vs. Sham group, *p* < 0.05, Figure 2C–E). To determine the best efficacy of puerarin, three doses (20 mg/kg, 40 mg/kg, and 80 mg/kg) were administered at 1 h after SAH [30,31]. Beam balance and modified Garcia tests were used to evaluate neurological functions of rats at 24 h after SAH. As shown in Figure 2F,G, there were significant neurological impairments in all SAH groups at 24 h after SAH when compared with the Sham group (*p* < 0.05). The modified Garcia test and beam balance test showed that puerarin at doses of 40 mg/kg and 80 mg/kg significantly improved neurobehavioral performance when compared with vehicle group (*p* < 0.05, Figure 2F,G). Thus, the 40 mg/kg was used as the optimal dose of puerarin in the subsequent studies.

### 3.3. Puerarin Treatment Improved Ipsilateral Hemisphere Oxidative Stress Injury at 24 h after SAH

Western blot showed that the oxidative stress marker 4HNE and antioxidative stress marker HO1 in the left ipsilateral hemisphere were significantly increased in the SAH + Vehicle group compared with Sham group at 24 h after SAH (*p* < 0.05, Figure 3A–C). Puerarin treatment further increased the protein level of HO1 and reduced the elevation of 4HNE in SAH rats when compared with the SAH + Vehicle group at 24 h after SAH (Figure 3A–C).

ELISA assay showed that the MDA level was significantly increased within the ipsilateral hemisphere, whereas SOD was significantly decreased in the SAH + Vehicle group compared with the Sham group at 24 h after SAH (Figure 3D,E). Puerarin treatment significantly reduced the levels of MDA, but further preserved the SOD level in the SAH + Puerarin group when compared with the SAH + Vehicle group (*p* < 0.05, Figure 3D,E).

8OHdG and Mitosox staining were used to assess DNA damage and mitochondrial oxidative-stress levels in ipsilateral brain tissue at 24 h after SAH. The numbers of 8OHdG- and mitosox-positive neurons were significantly increased in the SAH + Vehicle group compared with the Sham group (*p* < 0.05, Figure 3F). Such increases were significantly minimized by puerarin treatment in the SAH + puerarin group (*p* < 0.05, Figure 3F).

### 3.4. Puerarin Treatment Reduced Ferroptosis and Neuronal Degeneration at 24 h after SAH

Western blot showed that expression of the pro-ferroptosis protein, ACSL4, was significantly elevated, but the anti-ferroptosis protein, GPX4, was significantly decreased after SAH. Puerarin treatment significantly reduced ACSL4 levels and further increased the expression of GPX4 when compared with the vehicle-treated SAH rats (*p* < 0.05, Figure 4A–C) at 24 h after SAH.

ELISA showed that the GSH level was significantly decreased within the ipsilateral hemisphere, but the GSSG expression was significantly increased in the SAH + Vehicle group when compared with the Sham group at 24 h after SAH (Figure 4D,E). Puerarin treatment significantly increased the levels of GSH, but further reduced the expression of GSSG in the SAH + Puerarin group compared with the SAH + Vehicle group (*p* < 0.05, Figure 4D,E).

Brain iron levels were significantly increased in vehicle-treated SAH rats compared with the Sham group at 24 h after SAH (*p* < 0.05, Figure 4F). Puerarin treatment significantly reduced the levels of brain tissue iron in the SAH + Puerarin group when compared with the SAH + Vehicle group (*p* < 0.05, Figure 4F).

TUNEL staining was performed to assess for neuronal degeneration and apoptosis. Greater numbers of TUNEL-positive neurons were found in the SAH + Vehicle group when compared with the Sham group at 24 h after SAH. Consistently, the neuronal damage was significantly improved after puerarin treatment at the dose of 40 mg/kg when compared with vehicle-treated rats at 24 h after SAH (*p* < 0.05, Figure 4G).

### 3.5. Puerarin Treatment Improved Long-Term Neurological Deficits and Reduced Neuronal Degeneration at 26 Days after SAH

In Rotarod tests of 5 and 10 RPMs, the results showed that SAH rats had significantly shorter falling latency than Sham group at 1, 2, and 3 weeks after SAH. (*p* < 0.01, Figure 5A,B). The falling latency on both 5 and 10 RPMs were significantly improved with puerarin treatment in the SAH + Puerarin group compared with the SAH + Vehicle group (*p* < 0.01, Figure 5A,B).

In the Morris water maze test, the SAH + Vehicle group had an increased swim distance, increased escape latency to reach the platform, and less time in the platform quadrant when compared with the Sham group (*p* < 0.05, Figure 5C–E). When compared with the vehicle group, the puerarin-treated SAH rats had significantly improved spatial-learning memory, including reduced swimming distance and platform search time, as well as increased time spent in the target quadrant in probe trial (*p* < 0.05, Figure 5C–F). No significant differences were found in the swimming speed among all three groups (*p* > 0.05, Figure 5G).

Nissl staining showed that the neurons presented as shrunken cell bodies and condensed nuclei at 26 days after SAH. The puerarin treatment significantly attenuated neuronal injury in the CA1 region of the ipsilateral hippocampus compared to the SAH + Vehicle group (*p* < 0.05, Figure 5H,I).

### 3.6. Inhibition of AMPK with Dorsomorphin Abolished the Antioxidative and Anti-Ferroptosis Effects of Puerarin

To verify the neuroprotective mechanisms of puerarin treatment, the selective AMPK inhibitor, dorsomorphin, was administered via i.c.v. injection at 1 h before SAH. The modified Garcia and beam balance tests showed that the use of dorsomorphin abolished the neuroprotective effects of puerarin after SAH when compared with the SAH + Puerarin + Vehicle 2 group (*p* < 0.05; Figure 6A,B). The ELISA assay showed that the levels of MDA and GSSG were significantly increased within the ipsilateral hemisphere, whereas SOD and GSH expressions were significantly decreased in the SAH + Puerarin + dorsomorphin group compared with the Vehicle 2 group at 24 h after SAH (*p* < 0.05, Figure 6C–F). Western blotting showed that the expression of pAMPK, PGC1α, Nrf2, 4HNE, HO1, and ACSL4 was significantly increased, but the level of GPX4 was decreased at 24 h after SAH in the SAH + Vehicle 1 group compared to the Sham group (*p* < 0.05, Figure 6G,H). Moreover, puerarin treatment further increased the protein levels of pAMPK, PGC1α, Nrf2, HO1, and GPX4, but decreased the levels of 4HNE and ACSL4 when compared with the Vehicle 1-treated SAH group. However, inhibition of AMPK with dorsomorphin significantly decreased the levels of pAMPK, PGC1α, Nrf2, HO1, and GPX4, but increased the levels of 4HNE and ACSL4 in the SAH + Puerarin + Dorsomorphin group compared to the SAH + Puerarin + Vehicle 2 group (*p* < 0.05, Figure 6G,H).

### 3.7. Inhibition of PGC1α with SR18292 Abolished the Antioxidative and Anti-Ferroptosis Effects of Puerarin

A PGC1α inhibitor, SR18292, was used to further verify the signaling pathway involved in the neuroprotection of puerarin. The results showed that the SR18292 reversed the neurobehavioral benefits of puerarin after SAH compared to the SAH + Puerarin + Vehicle 3 group (*p* < 0.05, Figure 6A,B). ELISA assays showed that the levels of MDA and GSSG were significantly increased within the ipsilateral hemisphere, whereas SOD and GSH expressions were significantly decreased in the SAH + Puerarin + SR18292 group compared with the Vehicle 3 group at 24 h after SAH (*p* < 0.05, Figure 6C–F). Western blots showed that SR18292 administration significantly decreased the levels of PGC1α, Nrf2, HO1, and GPX4, but increased the levels of 4HNE and ACSL4 in the SAH + Puerarin + SR18292 group when compared to the SAH + Puerarin + Vehicle 3 group at 24 h after SAH (*p* < 0.05, Figure 7A–H).

## 4. Discussion

The goal of this study was to demonstrate that puerarin attenuates oxidative stress and ferroptosis of EBI partly through the AMPK/PGC1α/Nrf2-signaling pathway after SAH. The results showed that: (1) the endogenous protein levels of pAMPK and PGC1α were increased, peaking at 24 h after SAH. (2) Puerarin treatment attenuated oxidative stress and ferroptosis in SAH rats, and significantly improved the neurological deficits and neuronal degeneration at both 24 h (short term) and 26 d (long term) after SAH. (3) Mechanistically, puerarin treatment increased the protein levels of pAMPK, PGC1α, Nrf2, HO1, SOD, GPX4, and GSH, but decreased the levels of 4HNE, MDA, ACSL4, GSSG, and iron in the ipsilateral hemisphere at 24 h after SAH. (4) The selective AMPK inhibitor dorsomorphin or PGC1α inhibitor SR18292 partially abolished the beneficial effects of puerarin on neurological dysfunctions, oxidative stress injury, and ferroptosis. Taken together, our results indicated that puerarin attenuated oxidative stress and ferroptosis, as well as improved the neurobehavioral impairments after SAH through the activation of the AMPK/PGC1α/Nrf2-signaling pathway (Figure 8).

Oxidative stress injury is one of the main causes of EBI after SAH [37]. When blood enters the subarachnoid space, blood cells rupture and release a large amount of hemoglobin and oxygen free radical products [38], causing local oxidative stress, leading to DNA damage, iron accumulation, and lipid peroxidation [39,40]. Ferroptosis is a novel form of cell death, which is characterized by the iron-dependent oxidative destruction of cellular membranes after the body’s antioxidant system is severely damaged [6]. Accumulating evidence has indicated that iron accumulation and lipid peroxidation play important roles in regulating the process of ferroptosis [41]. The brain iron level was overloaded after intracranial hemorrhage, including intracerebral hemorrhage, intraventricular hemorrhage, and SAH [40]. Puerarin was suggested to be anti-oxidative-stress and anti-ferroptosis in multiple diseases [24,42,43]. Song et al. [44] showed that puerarin exerted a protective role in ferroptosis through regulating iron-handling proteins and enhancing antioxidant capacity in iron-overload-induced retinal injury. Liu et al. [16] supported that puerarin can markedly restore SOD, CAT, and GPX activities, the GSH/GSSG ratio, and suppress DNA oxidative damage in the liver of lead-treated rats. Liu et al.’s study showed that puerarin reduced iron content and increased ROS elimination in the rat heart failure disease model [23]. In the present study, the results showed that the ROS productions, including MDA, iron accumulation, and DNA oxidative damage, were significantly increased, whereas antioxidant enzymes were strongly decreased within the ipsilateral hemisphere at 24 h after SAH. Early treatment with puerarin can effectively reduce oxidative stress damage and iron death after SAH. Previous studies showed that puerarin attenuated learning and memory impairments, as well as inhibiting oxidative-stress damage in an Alzheimer’s disease model [45] and chronic alcohol poisoning model [46]. Our study also found similar results. Puerarin treatment can significantly improve the short-term neurological scores and the long-term water maze memory ability of rats after SAH.

As modulators of intracellular energy status, both AMPK and PGC1α are extremely active after brain injury [47]. Gao et al. [48] found that AMPK/PGC1α pathway activation protected mitochondrial biosynthesis and function in cerebral ischemic stroke, while inhibition of this pathway can aggravate cerebral ischemia/reperfusion injury [49]. Chen et al. [50] showed that the phosphorylation of AMPK was strongly increased after intracerebral hemorrhage, and that regulating the AMPK-signaling pathway can effectively reduce neuroinflammatory damage after intracerebral hemorrhage in a mouse model. Huang et al. [51] showed that promoting AMPK phosphorylation and upregulation of PGC1α expression had a neuroprotective role by attenuating oxidative stress and neuronal apoptosis in a hypoxic–ischemic encephalopathy rat model. As a master antioxidant regulator, Nrf2 was regulated by the AMPK/PGC1α-signaling pathway [52,53]. Recent studies have reported that activation of AMPK exerted a protective effect in ferroptosis-associated renal ischemia/reperfusion injury [20]. Additionally, Nrf2 was also shown to play a critical role in mitigating lipid peroxidation and ferroptosis in neurodegenerative diseases [54]. In the current study, the results showed that expression of pAMPK and PGC1α were increased at 3 h and peaked at 24 h after SAH, suggesting the activation of endogenous neuroprotective mechanisms after SAH. However, such activations appear not to be sufficient to overcome the SAH pathology. Puerarin treatment further increased brain expression of pAMPK, PGC1α, and Nrf2, but reduced the brain expression of oxidative-stress- and ferroptosis-related proteins at 24 h after SAH. Mechanistically, either the AMPK inhibitor or PGC1α inhibitor reversed the neurobehavioral benefits and anti-oxidative-stress and anti-ferroptosis effects of puerarin treatment. However, it is not clear whether puerarin directly activated AMPK or a certain receptor. Nevertheless, the results suggested that puerarin attenuated oxidative stress and ferroptosis through the AMPK/PGC1α/Nrf2-signaling pathway after SAH in the rat model.

There were several limitations in this study. First, in the current study, we characterized the anti-oxidative-stress and anti-ferroptosis effects of puerarin against EBI after SAH. However, other studies have reported the neuroprotective role of puerarin in regulating inflammation [55] and anti-apoptosis [56]. Therefore, further investigations are required to focus on the anti-neuroinflammatory and anti-neuronal apoptotic effects of puerarin after SAH. Second, puerarin was shown to play a neuroprotective role by activating multiple signaling pathways, demonstrating that other signaling pathways other than AMPK/PGC1α/Nrf2 warrant further exploration in future SAH studies. Third, dorsomorphin is an inhibitor of AMPK and ALK2/3/6 and is involved in BMP/SMAD signaling and the regulation of hepcidin expression in iron homeostasis. This signaling pathway should be focused on in future SAH studies. Fourth, only the iron concentration was analyzed in this study, and the iron-related proteins measured would be more beneficial to assess the presence of ferroptosis after SAH. Fifth, the number of rats used for immunofluorescence and Nissl staining is relatively small and further validation is needed in future studies. Sixth, considering that estrogen may affect the accuracy of the effects of puerarin after SAH, we only conducted the study in male rats [57]. Therefore, the neuroprotective effects in female rats need to be further elucidated after SAH. Seventh, the dose and time point of puerarin treatment chosen in the present study were based on the published literature. It is necessary to further optimize the concentration and time windows of puerarin treatment for treating SAH.

## 5. Conclusions

In conclusion, the results of this study indicated that puerarin treatment exerted neuroprotective effects against EBI after SAH by attenuating oxidative stress and ferroptosis in rats. The anti-oxidative and anti-ferroptotic effects were exerted partly through activating the AMPK/PGC1α/Nrf2 pathway. Thus, puerarin may serve as a promising therapeutic for the management of EBI in SAH patients.

## Figures and Tables

**Figure 1 antioxidants-11-01259-f001:**
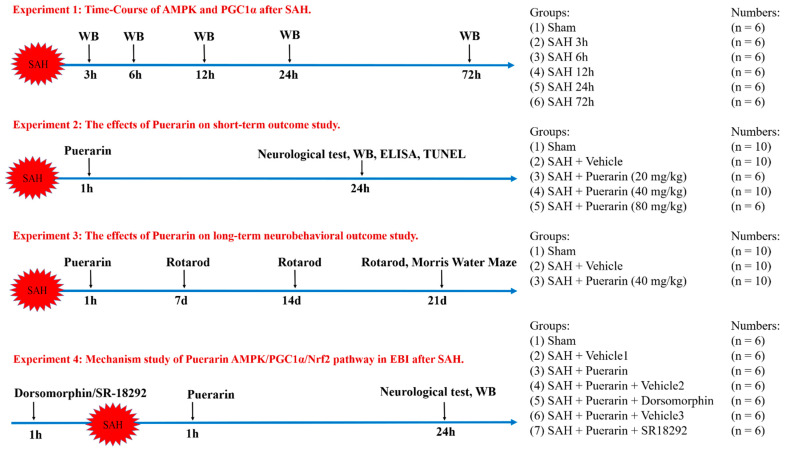
Experimental design and animal groups.

**Figure 2 antioxidants-11-01259-f002:**
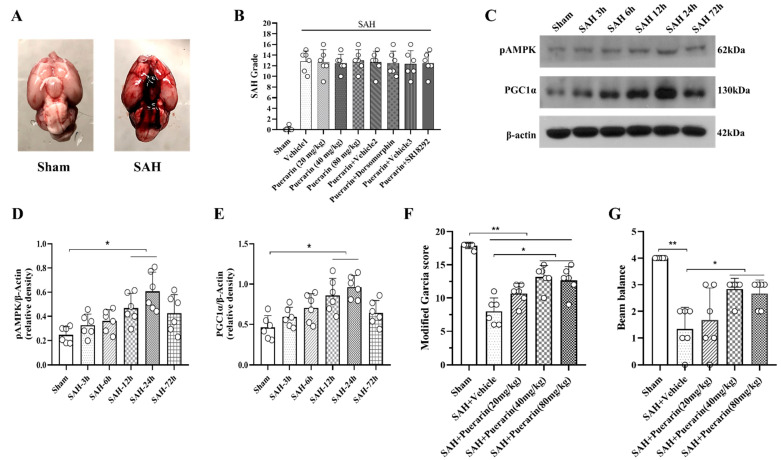
The temporal expression of pAMPK and PGC1α within ipsilateral brain hemisphere after subarachnoid hemorrhage (SAH). (**A**) Representative pictures showed that subarachnoid blood clots mainly presented around the circle of Willis in the rat brain at 24 h after SAH; (**B**) SAH grading scores of all SAH groups. (**C**) Representative Western blot bands of pAMPK and PGC1α; (**D**) Time course and densitometric quantification of pAMPK. (**E**) Time course and densitometric quantification of PGC1α. (**F**) Modified Garcia score at 24 h after SAH; (**G**) Beam balance score at 24 h after SAH. Data were represented as mean ± SD, * *p* < 0.05, ** *p* < 0.01, *n* = 6 per group, one-way ANOVA Tukey test was used for the comparison between sham and SAH groups.

**Figure 3 antioxidants-11-01259-f003:**
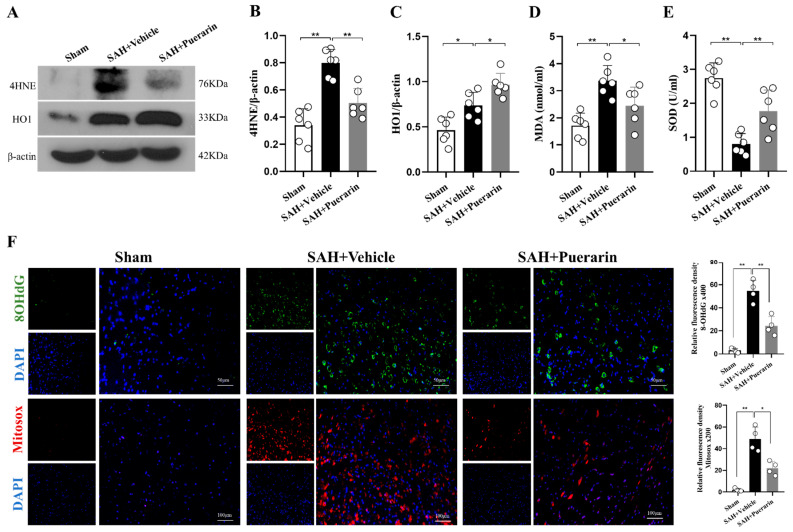
Puerarin treatment attenuated oxidative stress at 24 h after SAH. (**A**) Representative Western blot bands of 4HNE and HO1 at 24 h after SAH; (**B**,**C**) Densitometric quantification for 4HNE and HO1 expressions at 24 h after SAH; (**D**,**E**) Relative MDA and SOD level in ipsilateral brain hemisphere at 24 h after SAH; (**F**) Representative microphotographs and quantitative analysis for 8-OHdG and mitosox staining in the ipsilateral basal cortex. Scale bar = 50 or 100 μm. * *p* < 0.05, ** *p* < 0.01. Data are represented as mean ± SD, *n* = 4 or 6 per group, one-way ANOVA Tukey test.

**Figure 4 antioxidants-11-01259-f004:**
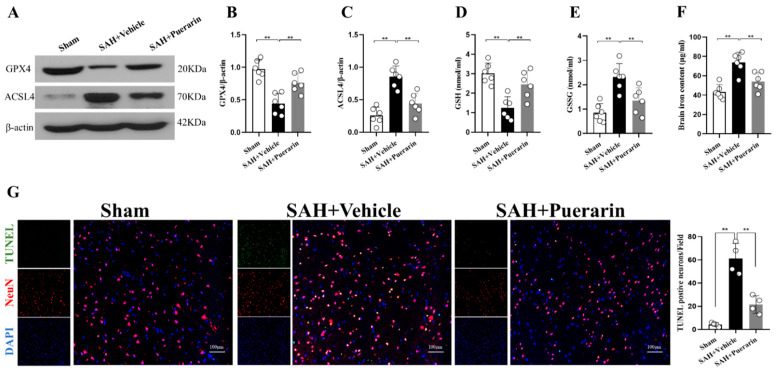
Puerarin treatment attenuated the ferroptosis and neuronal damage at 24 h after SAH. (**A**) Representative Western blot bands of GPX4 and ACSL4 at 24 h after SAH; (**B**,**C**) Densitometric quantification for GPX4 and ACSL4 expressions in brain at 24 h after SAH; (**D**,**E**) Relative GSH and GSSG level in ipsilateral brain hemisphere at 24 h after SAH; (**F**) Brain iron concentration in ipsilateral hemisphere at 24 h after SAH; (**G**) Representative microphotographs and quantitative analysis for TUNEL staining in the ipsilateral cortex of rat brain. Scale bar = 100 μm. ** *p* < 0.01. Data are represented as mean ± SD, *n* = 4 per group, one-way ANOVA Tukey test.

**Figure 5 antioxidants-11-01259-f005:**
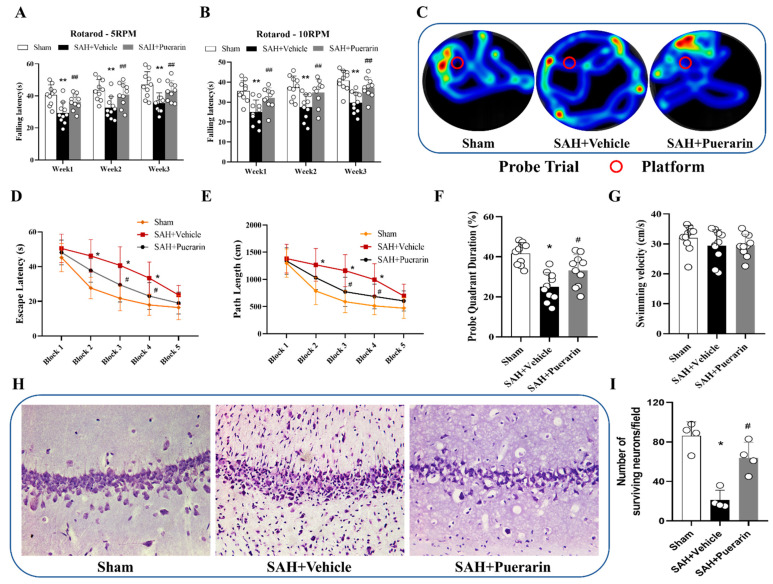
Puerarin treatment improved long-term outcomes after SAH. (**A**,**B**) Rotarod tests of 5 rpm and 10 rpm at every week after SAH; (**C**) Representative heat map of probe in Morris water maze test at day 26 after SAH; (**D**,**E**) Escape latency and swim distance on days 21–26 after SAH; (**F**) Quantification of the probe quadrant duration in the probe trial at day 26 after SAH; (**G**) Swimming velocities of different groups in probe trial; (**H**) Representative microphotographs of Nissl staining within the hippocampal CA1 region showed the damaged neurons; (**I**) Quantifications of the surviving neuron within hippocampal CA1 region, * *p* < 0.05, ** *p* < 0.01 vs. Sham group; # *p* < 0.05, ## *p* < 0.01 vs. SAH + Vehicle group. Data are represented as mean ± SD, *n* = 10 per group for behavioral tests, *n* = 4 per group for Nissl staining. Two-way ANOVA-Tukey for repeated measures of Rotarod and water mazes. One-way ANOVA–Tukey for measurements in probe trial and Nissl staining.

**Figure 6 antioxidants-11-01259-f006:**
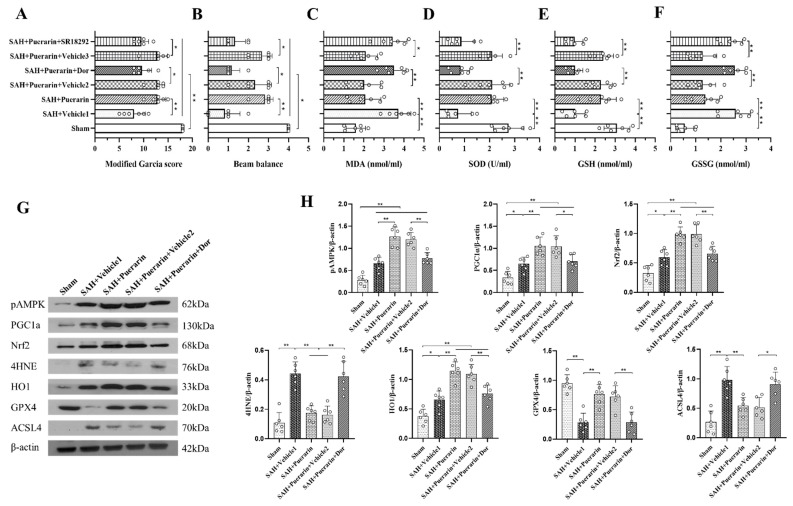
Dorsomorphin abolished the antioxidative stress and anti-ferroptosis effects of puerarin at 24 h after SAH. (**A**,**B**) AMPK inhibitor dorsomorphin partially reversed the neurobehavioral benefits of puerarin treatment at 24 h after SAH; (**C**–**F**) Relative MDA, SOD, GSH, and GSSG levels in ipsilateral brain hemisphere at 24 h after SAH; (**G**) Representative Western blot bands; (**H**) Densitometric quantification of pAMPK, PGC1α, Nrf2, 4HNE, HO1, GPX4, and ACSL4 in the ipsilateral brain hemisphere at 24 h after SAH. * *p* < 0.05, ** *p* < 0.01. Data are represented as mean ± SD, *n* = 6 per group. One-way ANOVA–Tukey.

**Figure 7 antioxidants-11-01259-f007:**
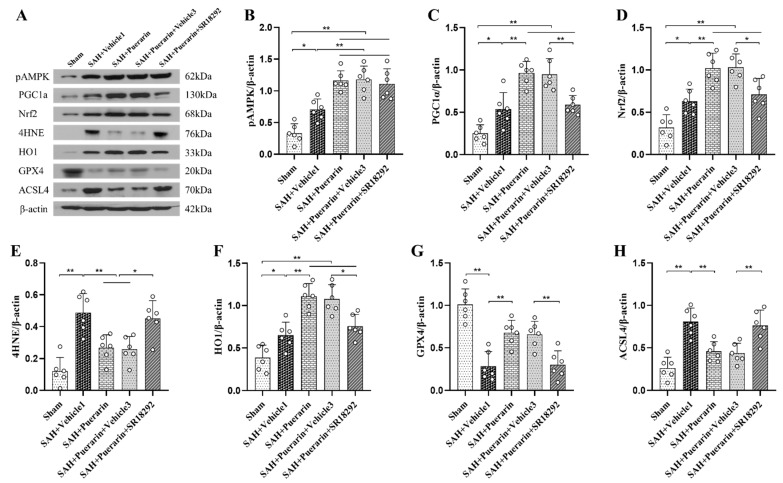
SR18292 abolished the anti-oxidative-stress and anti-ferroptosis effects of puerarin at 24 h after SAH. (**A**) Representative western blot bands; (**B**–**H**) Densitometric quantification of pAMPK, PGC1α, Nrf2, 4HNE, HO1, GPX4, and ACSL4 in the ipsilateral brain hemisphere at 24 h after SAH. * *p* < 0.05, ** *p* < 0.01. Data are represented as mean ± SD, *n* = 6 per group. One-way ANOVA–Tukey.

**Figure 8 antioxidants-11-01259-f008:**
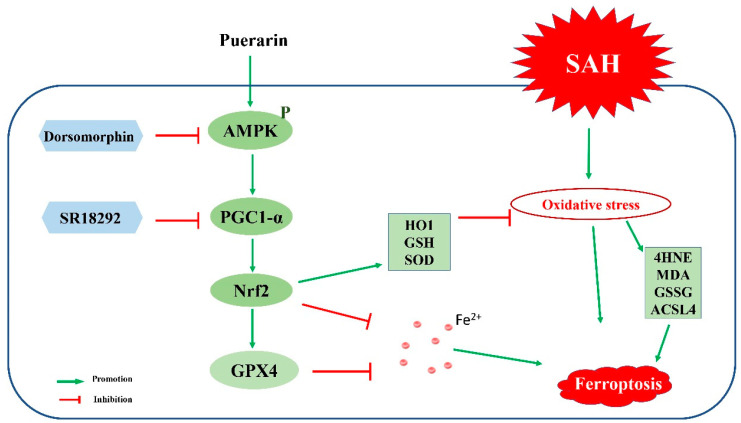
The graphic abstract. Puerarin attenuated oxidative stress and ferroptosis via activation of AMPK/PGC1α /Nrf2-signaling pathway after SAH.

**Table 1 antioxidants-11-01259-t001:** Summary of animal numbers and mortality.

Groups	Mortality	Excluded
Part 1: Time Course study		
Sham (*n* = 6) *	0.0% (0/6)	0
SAH (3 h, 6 h, 12 h, 24 h, 72 h) (*n* = 30) *	11.4% (4/35)	1
Part 2: Effects of Puerarin on short-term outcome study		
Sham (*n* = 10) *	0.0% (0/10)	0
SAH + Vehicle (*n* = 10) *	16.7% (2/12)	0
SAH + Puerarin (20 mg/kg) (*n* = 6)	12.5% (1/8)	1
SAH + Puerarin (40 mg/kg) (*n* = 10)	16.7% (2/12)	0
SAH + Puerarin (80 mg/kg) (*n* = 6)	22.2% (2/9)	1
Part 3: Effects of Puerarin on long-term outcome study		
Sham (*n* = 10)	0.0% (0/10)	0
SAH + Vehicle (*n* = 10)	16.7% (2/12)	0
SAH + Puerarin (*n* = 10)	16.7% (2/12)	0
Part 4: Mechanism study of Puerarin AMPK/PGC1α/Nrf2 pathway in EBI after SAH		
Sham (*n* = 6) *	0.0% (0/6)	0
SAH + Vehicle1 (*n* = 6) *	25.0% (2/8)	0
SAH + Puerarin (*n* = 6) *	14.3% (1/7)	0
SAH + Puerarin + Vehicle2 (*n* = 6)	14.3% (1/7)	0
SAH + Puerarin + Dorsomorphin (*n* = 6)	12.5% (1/8)	1
SAH + Puerarin + Vehicle3 (*n* = 6)	14.3% (1/7)	0
SAH + Puerarin + SR-1892 (*n* = 6)	12.5% (1/8)	1
Total		
Sham	0.0% (0/32)	0
SAH	15.2% (22/145)	5

* Shared groups.

## Data Availability

Data are contained within the article.

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
