# Peer review of "Puerarin Attenuates Oxidative Stress and Ferroptosis via AMPK/PGC1α/Nrf2 Pathway after Subarachnoid Hemorrhage in Rats"

_antioxidants, 2022, doi:10.3390/antiox11071259_

Round 1

Reviewer 1 Report

To the author:

The manuscript by Yi Huang et al. describes the neuroprotection effects of puerarin, a flavonoid, on early brain injury (EBI) in an animal model of subarachnoid hemorrhage (SAH) in rats. The study included behavioral studies and biochemical determinations for oxidative stress. The effect of puerarin on ferroptosis has been studied, but not in EBI. Thus, this manuscript seems very interesting in this field.

English is well-written.

In addition, there are some concerns that may be fixed, mainly related with the methodology:

-   What sex of animals was used? Male or females? In the case of only one sex, Why?

-   What volume of puerarin was used in its intranasal administration?

-   What was the volume of drugs used? 10 µl??

-   Please, specify how the SAH grade score is done. What is a score of 8? And a score of 9? How do you calculate? Although something is written in the point 2.5, it is not clear. Please specify what the modified Garcia score and Beam balance consist of.

-    Point 2.9. and Point 2.10. What brain areas were analyzed? Or was a homogenate of the whole brain?

-       Point 2.12. Regarding statistical analysis, please include which statistical methods were used for the assessment of normality and homoscedasticity.

-       Point 3.2. Please include the 3 doses of puerarin to evaluate in the point 2.3

-       Figure 6 should be horizontal for a better visualization of the figure. The font size is small, so placing it horizontally might allow be bigger enough.

-       Also of concern is the number of animals included in the study. The number of animals is low in some studies. It would be appropriate to include this as a limitation in the manuscript.

-        

Author Response

Response to review for (No. antioxidants-1716173): Puerarin attenuates oxidative stress and ferroptosis via AMPK/PGC1α/Nrf2 pathway after subarachnoid hemorrhage in rats

Dear Editors:

Thank you for your thoughtful review of our manuscript and for the opportunity to further improve our work by revision. We have made all of the required changes according to the suggestions of the reviewers. Please refer to our point-by-point response to all the inquiries below.

Reviewer #1:

The manuscript by Yi Huang et al. describes the neuroprotection effects of puerarin, a flavonoid, on early brain injury (EBI) in an animal model of subarachnoid hemorrhage (SAH) in rats. The study included behavioral studies and biochemical determinations for oxidative stress. The effect of puerarin on ferroptosis has been studied, but not in EBI. Thus, this manuscript seems very interesting in this field.

English is well-written.

In addition, there are some concerns that may be fixed, mainly related with the methodology:

-   What sex of animals was used? Male or females? In the case of only one sex, Why?

Response: Thank you for your suggestion. Previous studies have shown a protective effect of estrogen against EBI after SAH (Biol Rev Camb Philos Soc. 2019 Dec;94(6):1897-1917; Stroke. 2021 Aug;52(8):2661-2670). Female rat models may affect the accuracy of study results. Therefore, to avoid the effect of estrogen, in studies of EBI after SAH, investigators have routinely selected only male rats for testing. (Brain Behav Immun. 2021 Jan;91:587-600; Redox Biol. 2019 Jan;20:75-86.)

-   What volume of puerarin was used in its intranasal administration? And What was the volume of drugs used? 10 µl??

Response: Thank you for your suggestion. The drugs were administered via intranasal route with 20 µl, and we have added a detailed description in the part of 2.3 Drug administration.

“Briefly, rats were anesthetized and placed in the supine position. A total volume of 20 µl of vehicle or three different concentrations of puerarin (20mg/kg, 40 mg/kg and 80 mg/kg) were slowly injected with 5 μl/one nostril and alternated every 5 min.”

-   Please, specify how the SAH grade score is done. What is a score of 8? And a score of 9? How do you calculate? Although something is written in the point 2.5, it is not clear. Please specify what the modified Garcia score and Beam balance consist of.

Response: Thank you for your suggestion. We have added the SAH grade calculation in the part of 2.2 SAH model, and modified Garcia score and Beam balance in the part of 2.5 Neurological score.

“Referring to Sugawara's study [26], the brain was organized into 6 regions and scored from 0-3 according to the amount of blood covering the surface of each region, and the total SAH grade score was the sum of the scores for each region (0-18).”

“The neurological scores were assessed using the Modified Garcia test and beam balance test [25]. The Modified Garcia score (3-18) was the sum of 6 separate trial scores, including spontaneous activity (scores 0-3), symmetry (0-3), forelimb extension (0-3), climbing (1-3), body proprioception (1-3) and response to vibrators (1-3). Beam balance test was used to record the distance rats walked on a balance beam in a quiet environment, scoring 0-4.”

-    Point 2.9. and Point 2.10. What brain areas were analyzed? Or was a homogenate of the whole brain?

Response: Thank you for your suggestion. The ipsilateral brain tissues from rat SAH model were used for the western blot analysis and ELISA assay. We have added that in the Point 2.9. and Point 2.10.

-       Point 2.12. Regarding statistical analysis, please include which statistical methods were used for the assessment of normality and homoscedasticity.

Response: Thank you for your suggestion. We have added that in the Point 2.12. “Normality and homoscedasticity assumptions were checked with a Kolmogorov–Smirnov test. ”

-       Point 3.2. Please include the 3 doses of puerarin to evaluate in the point 2.3

Response: Thank you for your suggestion. We have revised this in the article.

“Briefly, rats were anesthetized and placed in the supine position. A total volume of 20 µl of vehicle or three different concentrations of puerarin (20mg/kg, 40 mg/kg and 80 mg/kg) were slowly injected with 5 μl/one nostril and alternated every 5 min. ”

-       Figure 6 should be horizontal for a better visualization of the figure. The font size is small, so placing it horizontally might allow be bigger enough.

Response: Thank you for your suggestion. We have revised this in the article.

-       Also of concern is the number of animals included in the study. The number of animals is low in some studies. It would be appropriate to include this as a limitation in the manuscript.

Response: Thank you for your suggestion. We have revised this in the article.

“There were several limitations in this study. First, in the current study, we charac-terized the anti-oxidative stress and anti-ferroptosis effects of puerarin against EBI af-ter SAH. However, other studies have reported the neuroprotective role of puerarin in regulating inflammation [55] and anti-apoptosis [56]. Therefore, further investigations are required to focus on the anti-neuroinflammatory and anti-neuronal apoptotic ef-fects of puerarin after SAH. Second, puerarin was shown to play a neuroprotective role by activating multiple signaling pathways, demonstrating that other signaling path-ways other than AMPK/PGC1α/Nrf2 warrant further exploration in future SAH stud-ies. Third, Dorsomorphin is an inhibitor of AMPK and ALK2/3/6 and is involved in BMP/SMAD signaling and regulation of hepcidin expression in iron homeostasis. This signaling pathway should be focused on in future SAH studies. Fourth, only the iron concentration was analyzed in this study, and the iron-related proteins measured would be more beneficial to assess the presence of ferroptosis after SAH. Fifth, the number of rats used for immunofluorescence and Nissl staining is relatively small and further validation is needed in future studies. Sixth, the dose and time point of puerarin treatment chosen in the present study were based on the published literatures. It is necessary to further optimize the concentration and time windows of puerarin treatment for treating SAH.”

Reviewer 2 Report

Antioxidants

Puerarin Attenuates Oxidative Stress and Ferroptosis via AMPK/PGC1/Nrf2 Pathway after Subarachnoid Hemorrhage in Rats

Yi Huang, Honggang Wu, Yongmei Hu, Chenhui Zhou, et al.

Overall Impression:  The study presented by Huang et al, is fairly impressive with a large amount of data that suggest the phytochemical puerarin has neuroprotective effects mediated in turn by its actions on oxidative stresss and ferroptosis.  There are some minor issues and one major issue as outlined below.

Specific Comments:

1.     A major concern is that only male rates were used in this study.  The NIH has strict guidelines that any study funded by their agency must use both genders, unless there is convincing reason for only using one gender.  I realize that this study is not funded by the NIH, but I would like the authors to explain why they only used male mice.

2.   Please list the purity and catalog number for Sigma puerarin.  Also the same information should be listed for all the other chemical listed in this study.  This is vital for any laboratory that wants to replicate the authors’ work.

3.   In Fig. 1, Why do some groups of rats have an n of 6,, while others have an n 10?

4.   Were power calculations performed to approximate the number of animals needed for each type of experiment?

5.   In Table 1.  It appears that mortality increases between 40-80 mg/kg of puerarin.  Is this correct?

6.   Based on the use of selective inhibitors, it appears that PGC1a plays a critical role in the action of puerarin.  I realize these were animal experiments, but perhaps in future studies siRNA could be used, as it has now in humans as a therapeutic, to more strongly prove that PGC1a is indeed the critical component.

7.   Lastly I would like to compliment the authors on the English in this manuscript.  It was one of the best manuscripts I have seen, in this regard, originating from a lab in China.

Author Response

Response to review for (No. antioxidants-1716173): Puerarin attenuates oxidative stress and ferroptosis via AMPK/PGC1α/Nrf2 pathway after subarachnoid hemorrhage in rats

Dear Editors:

Thank you for your thoughtful review of our manuscript and for the opportunity to further improve our work by revision. We have made all of the required changes according to the suggestions of the reviewers. Please refer to our point-by-point response to all the inquiries below.

Reviewer #2:

Puerarin Attenuates Oxidative Stress and Ferroptosis via AMPK/PGC1a /Nrf2 Pathway after Subarachnoid Hemorrhage in Rats

Yi Huang, Honggang Wu, Yongmei Hu, Chenhui Zhou, et al.

Overall Impression:  The study presented by Huang et al, is fairly impressive with a large amount of data that suggest the phytochemical puerarin has neuroprotective effects mediated in turn by its actions on oxidative stresss and ferroptosis.  There are some minor issues and one major issue as outlined below.

Specific Comments:

  1. A major concern is that only male rates were used in this study.  The NIH has strict guidelines that any study funded by their agency must use both genders, unless there is convincing reason for only using one gender.  I realize that this study is not funded by the NIH, but I would like the authors to explain why they only used male mice.

Response: Thank you for your suggestion. Previous studies have shown a protective effect of estrogen against EBI after SAH (Biol Rev Camb Philos Soc. 2019 Dec;94(6):1897-1917; Stroke. 2021 Aug;52(8):2661-2670). Female rat models may affect the accuracy of study results. Therefore, to avoid the effect of estrogen, in studies of EBI after SAH, investigators have routinely selected only male rats for testing. (Brain Behav Immun. 2021 Jan;91:587-600; Redox Biol. 2019 Jan;20:75-86.)

  1. Please list the purity and catalog number for Sigma puerarin.  Also, the same information should be listed for all the other chemical listed in this study.  This is vital for any laboratory that wants to replicate the authors’ work.

Response: Thank you for your suggestion. We have added the purity and catalog number in the article.

  1. In Fig. 1, Why do some groups of rats have an n of 6,, while others have an n 10? And, 4.   Were power calculations performed to approximate the number of animals needed for each type of experiment?

Response: Thank you for your suggestion. The power calculations were performed to approximate the number of animals needed for each type of experiment. So that the number of long-term outcome study was 10, and the number of short-term studies was 6. In addition, four more rats were used for immunofluorescence assay in SAH+PBS and SAH+Puerarin (40 mg/kg) groups, respectively.

  1. In Table 1.  It appears that mortality increases between 40-80 mg/kg of puerarin.  Is this correct?

Response: It is really true as Reviewer suggested. The current clinical mortality rate of SAH is very high, nearly 30%. the SAH model is also established with a high mortality rate. The authors are a neurosurgeon and have very skilled surgical techniques to effectively control their mortality. Table 1 is a very objective listing of the mortality rates of the rats in this experiment.

  1. Based on the use of selective inhibitors, it appears that PGC1a plays a critical role in the action of puerarin.  I realize these were animal experiments, but perhaps in future studies siRNA could be used, as it has now in humans as a therapeutic, to more strongly prove that PGC1a is indeed the critical component.

Response: Thank you for your suggestion. The reviewer has raised a very valuable point. PGC-1α has been shown to play a pivotal role in the regulation of mitochondrial biogenesis (Front Mol Biosci. 2021; 8: 620683.) and oxidative stress (Med Sci Monit. 2020; 26: e923688-1–e923688-14.) in EBI following SAH. Many investigators have also conducted studies in animal models with PGC1 inhibitors or siRNA and obtained effective results. Therefore, there is a great hope that PGC1 inhibitors or siRNAs can be translated into clinical practice in future studies (Theranostics 2021; 11(2):522-539.).

  1. Lastly I would like to compliment the authors on the English in this manuscript.  It was one of the best manuscripts I have seen, in this regard, originating from a lab in China.

Response: Thank you for your suggestion. We are honored to receive your approval for our paper.

Reviewer 3 Report

The work studies the effects of induced subarachnoid haemorrhage in a rat model and shows that it induces cell death, oxidative damage and the induction of AMPK/PGC1a/NrF2 pathway and that these effects are attenuated by previous treatment with puerarin. The work is carefully planned and well-presented, and the positive effects of puerarin are convincing. There are some problems in the interpretation of the results.

- In the title and throughout the text and discussion the authors state that the SAH treatment induces ferroptosis, probably because they observe oxidative damage and reduction of GX4 and GSH. I do not think that this is sufficient to assess that we are in the presence of ferroptosis for two reasons: first, ferroptosis is generally defined as the suppression of cell death by specific inhibitors, mainly ferrostatin 1, of by iron chelators, that were not attempted. Second, the TUNEL assay clearly showed the induction of apoptosis in the brain. Thus, I think the term ferroptosis should be removed from the title and discussed as a possibility in the text, alternatively data with specific inhibitors should be presented.

- Dorsomorphin is an inhibitor of AMPK and also of ALK2/3/6, involved in BMP/SMAD signalling and hepcidin expression, which regulates iron homeostasis. This should be discussed, and possibly the expression of hepcidin verified

- The incubation in 8.5 M HCl at 90°C is expected to release iron from heme, thus the procedure evaluated total iron, rather than nonheme iron. The analysis of ferritin would improve the assessment of iron status in the brain. 

Author Response

Reviewer #3:

The work studies the effects of induced subarachnoid haemorrhage in a rat model and shows that it induces cell death, oxidative damage and the induction of AMPK/PGC1a/NrF2 pathway and that these effects are attenuated by previous treatment with puerarin. The work is carefully planned and well-presented, and the positive effects of puerarin are convincing. There are some problems in the interpretation of the results.

- In the title and throughout the text and discussion the authors state that the SAH treatment induces ferroptosis, probably because they observe oxidative damage and reduction of GX4 and GSH. I do not think that this is sufficient to assess that we are in the presence of ferroptosis for two reasons: first, ferroptosis is generally defined as the suppression of cell death by specific inhibitors, mainly ferrostatin 1, of by iron chelators, that were not attempted. Second, the TUNEL assay clearly showed the induction of apoptosis in the brain. Thus, I think the term ferroptosis should be removed from the title and discussed as a possibility in the text, alternatively data with specific inhibitors should be presented.

Response: Thank you for your suggestion. We have made correction according to the Reviewer’s comments. The title of the paper was changed to "Puerarin attenuates oxidative stress via AMPK/PGC1α/Nrf2 pathway after subarachnoid hemorrhage in rats".

- Dorsomorphin is an inhibitor of AMPK and also of ALK2/3/6, involved in BMP/SMAD signalling and hepcidin expression, which regulates iron homeostasis. This should be discussed, and possibly the expression of hepcidin verified.

Response: Thank you for your suggestion. The reviewer has raised a very valuable point. We have added these in the limitations.

“Third, Dorsomorphin is an inhibitor of AMPK and ALK2/3/6 and is involved in BMP/SMAD signaling and regulation of hepcidin expression in iron homeostasis. This signaling pathway should be focused on in future SAH studies. Fourth, only the iron concentration was analyzed in this study, and the iron-related proteins measured would be more beneficial to assess the presence of ferroptosis after SAH. ”.

- The incubation in 8.5 M HCl at 90°C is expected to release iron from heme, thus the procedure evaluated total iron, rather than nonheme iron. The analysis of ferritin would improve the assessment of iron status in the brain. 

Response: Thank you for your suggestion. We have removed the word nonheme from the article.

The reviewer has raised a very valuable point. We added a discussion of ferritin in the limitations.

“Fourth, only the iron concentration was analyzed in this study, and the iron-related proteins measured would be more beneficial to assess the presence of ferroptosis after SAH. ”

Reviewer 4 Report

Huang et al. in their original article present results of study focusing on the impact of puerarin, flavonoid glycoside extracted from the Pueraria lobata root, on ferroptosis triggered by subarachnoid hemorrage (SAH) in rats. The authors cite the relevant literature items, which either constitute an appropriate introduction, and allow for a comprehensive discussion of the obtained results. However, it cannot be agreed that publication number 7 by Zhanng et al (2020) clearly indicates the role of ferroptosis in the EBI after SAH.

There remains only a small but significant problem with brain iron levels after SAH, shown in Figure 4F. What is the source of oxidative stress and the cause of lipid oxidation after SAH, hence the induction of ferropotosis? Free heme or maybe iron? As we know, free heme has tremendous pro-oxidative properties, but the apparent increase in HO1 (Fig. 6G) supposedly solves this problem. In this way, another free radical donor appears, namely iron (heme decay product), which generates HO in the Fenton reaction. A second question then arises, why does the iron level drop after Puerarin? What is the mechanism? Maybe this is where Puerarina's beneficial effect on the reduction of ferroptosis after SAH should be sought? I would be grateful for the discussion.

A final note concerns the Chinese company that produces ELISA tests. Unfortunately, the reviewers are not able to assess the quality or methodology of the research, as only Chinese manuals are available. Although they are quoted in many good scientific journals (for example for SOD). This makes it very difficult to make an honest review and respond to the point of review: Are the methods adequately described? So what was the principle of determining SOD activity?

The manuscript describes well-designed studies which proved beneficial effect of puerarin on SAH. The mechanism underlying this phenomenon proposed by the authors: activation of the AMPK / PGC1α / Nrf2 pathway is well supported by the results of the analyzes performed. The presented article brings valuable knowledge in this field and can be accepted for publication after cosmetic changes.

Based on the above, I believe that the current version of the manuscript meets the requirements of the Antioxidant journal and may be published prior to minor changes. Moreover, I believe that the conclusions reached by the authors are justified in the presented research results.

Author Response

Reviewer #4:

Huang et al. in their original article present results of study focusing on the impact of puerarin, flavonoid glycoside extracted from the Pueraria lobata root, on ferroptosis triggered by subarachnoid hemorrage (SAH) in rats. The authors cite the relevant literature items, which either constitute an appropriate introduction, and allow for a comprehensive discussion of the obtained results. However, it cannot be agreed that publication number 7 by Zhanng et al (2020) clearly indicates the role of ferroptosis in the EBI after SAH.

Response: Thank you for your suggestion. We have replaced reference 7.

There remains only a small but significant problem with brain iron levels after SAH, shown in Figure 4F. What is the source of oxidative stress and the cause of lipid oxidation after SAH, hence the induction of ferropotosis? Free heme or maybe iron? As we know, free heme has tremendous pro-oxidative properties, but the apparent increase in HO1 (Fig. 6G) supposedly solves this problem. In this way, another free radical donor appears, namely iron (heme decay product), which generates HO in the Fenton reaction. A second question then arises, why does the iron level drop after Puerarin? What is the mechanism? Maybe this is where Puerarina's beneficial effect on the reduction of ferroptosis after SAH should be sought? I would be grateful for the discussion.

Response: Thank you for your suggestion. Early brain injury after SAH was a very complex pathophysiological process. ROS were suggested as the key mediators of pathophysiological mechanisms involved in EBI after SAH.

Oxidative stress came from several sources in EBI after SAH. Mitochondrial disruption and subsequent endoplasmic reticulum stress played an important role in excessive production of ROS. Hemoglobin autoxidation and various interactions of OxyHb and MetHb also leaded ROS overproduction. Unregulated enzymatic pathways like xanthine dehydrogenase, NADPH oxidase, NOS, and derivates of arachidonic acid processing can also be a major source of ROS.

The current study suggested that iron overload was necessary for the iron death process. The blood degradation products in the subarachnoid space can lead to iron overload, which in turn activated lipid peroxidation and aggregation of reactive oxygen species, leading to early iron death after SAH.

Other studies showed that puerarin exerted a protective role in ferroptosis through regulating iron-handling proteins and enhancing antioxidant capacity in iron overload-induced retinal injury.

In this study, our results showed that the puerarin treatment significantly increased the protein levels of HO1, SOD, GPX4, and GSH, decreased the levels of 4HNE, MDA, ACSL4, GSSG, and iron through the activation of AMPK/PGC1α/Nrf2 signaling pathway in the SAH model. So that the AMPK/PGC1α/Nrf2 signaling pathway may be a key pathway for the protective effect of geranium in early brain injury after SAH.

Our results showed that puerarin increased the concentration of GPX4 and GSH, two proteins that reduce lipid peroxidation as well as the Fenton reaction, etc. Of course, ferritin plays an important role in this process, and our study lacked the analysis of ferritin, which we have discussed in the limitation.

A final note concerns the Chinese company that produces ELISA tests. Unfortunately, the reviewers are not able to assess the quality or methodology of the research, as only Chinese manuals are available. Although they are quoted in many good scientific journals (for example for SOD). This makes it very difficult to make an honest review and respond to the point of review: Are the methods adequately described? So what was the principle of determining SOD activity?

Response: Thank you for your suggestion. We have added detailed steps and the rationale for detecting brain oxidative damage and ferroptosis indexes in the Materials Methods.

“Briefly, MDA detection was based on the chromogenic reaction between MDA and thiobarbituric acid, and measured at 535 nm light wave [33]. SOD analysis was per-formed according to the WST-8 method and detected at 450 nm light wave [34, 35]. The GSH/GSSG assay was performed by glutathione reductase treatment and the concen-tration is detected at 412 nm light wave [33].

The manuscript describes well-designed studies which proved beneficial effect of puerarin on SAH. The mechanism underlying this phenomenon proposed by the authors: activation of the AMPK / PGC1α / Nrf2 pathway is well supported by the results of the analyzes performed. The presented article brings valuable knowledge in this field and can be accepted for publication after cosmetic changes.

Based on the above, I believe that the current version of the manuscript meets the requirements of the Antioxidant journal and may be published prior to minor changes. Moreover, I believe that the conclusions reached by the authors are justified in the presented research results.

Response: Thank you for your suggestion. We are honored to receive your approval for our paper.

Round 2

Reviewer 1 Report

To the author:

The manuscript by Yi Huang et al. describes the neuroprotection effects of puerarin, a flavonoid, on early brain injury (EBI) in an animal model of subarachnoid hemorrhage (SAH) in rats. The study included behavioral studies and biochemical determinations for oxidative stress. The effect of puerarin on ferroptosis has been studied, but not in EBI. Thus, this manuscript seems very interesting in this field.

Most of the answers are satisfactory however, there are still some concerns that may be fixed:

-   Regarding the sex of the animals, an explanation of the use of only males have to be included in the manuscript. Authors answered the question but they did not include anything in the manuscript.

-       Point 2.12. Regarding statistical analysis, please include which statistical methods were used for the assessment of homoscedasticity.

Author Response

The manuscript by Yi Huang et al. describes the neuroprotection effects of puerarin, a flavonoid, on early brain injury (EBI) in an animal model of subarachnoid hemorrhage (SAH) in rats. The study included behavioral studies and biochemical determinations for oxidative stress. The effect of puerarin on ferroptosis has been studied, but not in EBI. Thus, this manuscript seems very interesting in this field.

Most of the answers are satisfactory however, there are still some concerns that may be fixed:

-   Regarding the sex of the animals, an explanation of the use of only males have to be included in the manuscript. Authors answered the question but they did not include anything in the manuscript.

Response: Thank you for your suggestion. We have added that in the limitations.

“Sixth, considering that estrogen may affect the accuracy of the effects of puerarin after SAH, we only conducted the study in male rats [57]. Therefore, the neuroprotective effects in female rats need to be further elucidated after SAH.”

-       Point 2.12. Regarding statistical analysis, please include which statistical methods were used for the assessment of homoscedasticity.

Response: Thank you for your suggestion. I am very sorry for not understanding your meaning correctly, we have reworked it according to your questions.

All statistical analyses were performed using GraphPad Prism 8 software (CA, USA).  The protein, immunostaining, Nissl staining and neurological scores were firstly evaluated for the normality and homogeneity of variances using Shapiro–Wilk test and Levene’s test, respectively, and then were analyzed using one-way ANOVA followed by Tukey’s post-hoc test. Rotarod and water mazes tests were measured by two-way ANOVA Tukey’s post-hoc test. The results were presented as mean ± standard deviation (SD), and p < 0.05 was considered statistically significant.”

Reviewer 3 Report

The authors provided adequate answers to my points

Author Response

The authors provided adequate answers to my points.

Response: Thank you for your suggestion. We are honored to receive your approval for our paper.